# How Can We Best Measure Frailty in Cardiosurgical Patients?

**DOI:** 10.3390/jcm12083010

**Published:** 2023-04-20

**Authors:** Magdalena L. Laux, Christian Braun, Filip Schröter, Daniela Weber, Aiman Moldasheva, Tilman Grune, Roya Ostovar, Martin Hartrumpf, Johannes Maximilian Albes

**Affiliations:** 1Department of Cardiovascular Surgery, Heart Center Brandenburg, University Hospital Brandenburg Medical School, Faculty of Health Sciences Brandenburg, 16321 Bernau, Germany; 2Department of Molecular Toxicology, German Institute of Human Nutrition (DIfE), 14558 Nuthetal, Germany; 3Department of Biomedical Sciences, Nazarbayev University, Astana 010000, Kazakhstan

**Keywords:** cardiosurgery, frailty, risk score, outcome

## Abstract

Background: Frailty is gaining importance in cardiothoracic surgery and is a risk factor for adverse outcomes and mortality. Various frailty scores have since been developed, but there is no consensus which to use for cardiac surgery. Methods: In an all-comer prospective study of patients presenting for cardiac surgery, we assessed frailty and analyzed complication rates in hospital and 1-year mortality, as well as laboratory markers before and after surgery. Results: 246 included patients were analyzed. A total of 16 patients (6.5%) were frail, and 130 patients (52.85%) were pre-frail, summarized in the frail group (FRAIL) and compared to the non-frail patients (NON-FRAIL). The mean age was 66.5 ± 9.05 years, 21.14% female. The in-hospital mortality rate was 4.88% and the 1-year mortality rate was 6.1%. FRAIL patients stayed longer in hospital (FRAIL 15.53 ± 8.5 days vs. NON-FRAIL 13.71 ± 8.94 days, *p* = 0.004) and in intensive/intermediate care units (ITS/IMC) (FRAIL 5.4 ± 4.33 days vs. NON-FRAIL 4.86 ± 4.78 days, *p* = 0.014). The 6 min walk (6 MW) (317.92 ± 94.17 m vs. 387.08 ± 93.43 m, *p* = 0.006), mini mental status (MMS) (25.72 ± 4.36 vs. 27.71 ± 1.9, *p* = 0.048) and clinical frail scale (3.65 ± 1.32 vs. 2.82 ± 0.86, *p* = 0.005) scores differed between patients who died within the first year after surgery compared to those who survived this period. In-hospital stay correlated with timed up-and-go (TUG) (TAU: 0.094, *p* = 0.037), Barthel index (TAU-0.114, *p* = 0.032), hand grip strength (TAU-0.173, *p* < 0.001), and EuroSCORE II (TAU 0.119, *p* = 0.008). ICU/IMC stay duration correlated with TUG (TAU 0.186, *p* < 0.001), 6 MW (TAU-0.149, *p* = 0.002), and hand grip strength (TAU-0.22, *p* < 0.001). FRAIL patients had post-operatively altered levels of plasma-redox-biomarkers and fat-soluble micronutrients. Conclusions: frailty parameters with the highest predictive value as well as ease of use could be added to the EuroSCORE.

## 1. Introduction

Among cardiosurgical surgery patients in Germany, the percentage of the elderly is steadily increasing [1]. Therefore, more frail patients are seen in our daily practice, and it is our task as surgeons to evaluate the risk and benefit of an operation, as minimally invasive strategies or transcatheter approaches may be alternatives [2]. Various perioperative risk scores calculate perioperative risks, but cannot determine frailty, which leads to considerable heterogeneity in old patients [3,4,5,6]. EuroSCORE and STS scores are widely used to estimate perioperative risk [7]. The EuroSCORE seems to overestimate mortality related to lower scores and underestimate mortality related to higher scores [8] Some additionally describe an overestimation in high-risk patients [9]. The original EuroSCORE has thus been supplemented by the logistic EuroSCORE and the EuroSCORE II to at least partially compensate for these phenomena [6], and may still need to be adjusted for further risks not yet present in the current score [10]. The STS-Score in contrast is known to slightly underestimate perioperative risk [11]. Nevertheless, however, some individual aspects remain elusive in these scores although it is known that some patients are more vulnerable to the burden of surgery than others. This is reflected by the concept of a frail versus a spry condition [12,13]. Frailty was first described and classified by Linda Fried [14]. Various frailty scores have since been developed and compared [15,16,17,18]. Indeed, in cardiosurgical patients, different risk factors for frailty have shown to serve as additional predictors of outcome but the instruments used vary [19,20,21,22,23]. The meta-analysis by Lee et al. compared 19 studies that used the Fried/Modified Fried (25%), Deficit Index (17%), Bespoke Frailty Score (17%), Clinical Frailty Score (8%) and Katz Index (8%) and walking velocity (71%), 6-min walk test (14%) and psoas muscle measurement (14%) to screen for the presence of frailty [24].

Efforts are being made to identify appropriate biomarkers of frailty to obtain reasonably quantifiable parameters [25,26]. Inflammatory parameters, mitochondrial function, calcium-homeostasis, markers of fibrosis, neuronal impairment and markers of the cytoskeleton among others hane been investigated to correlate promisingly with the extent of frailty.

In the absence of a standardized frailty measure, we aimed to compare a number of different assessments of frailty in cardiosurgical patients in order to use a highly predictive but time- and cost-effective score.

## 2. Materials and Methods

In this prospective observational study called PREDARF (PRE-operative Detection of Age-Related Factors; funded by the Brandenburg State Ministry of Science), we included all elective patients presenting for cardiac surgery. In the “all-comers” concept from December 2017 to May 2019, we therefore excluded emergency patients such as those with acute aortic dissection, acute myocardial infarction, and active endocarditis. After informed consent, patients were screened for frailty symptoms according to Fried (weight loss, hand grip strength, fatigue, 5 m walking test, and weekly activities). All frail (3 or more criteria met) and pre-frail (1 or 2 criteria met) patients were evaluated as the frailty group (FRAIL) and compared to the non-frail patients (NON-FRAIL). As frail patients tend to be referred for a less invasive procedure, and they might not be drawn to a study focusing on mobility testing, in our cohort who underwent cardiac surgery, the portion of frail patients was low. We had a big portion of pre-frail patients, as described in the results section. Therefore, for the statistical analysis, we formed two groups. Follow-up was conducted before discharge in our department and also via telephone questionnaire 4 to 6 months and 1 year after surgery. The study was approved by the ethics committee (no. S19(a)/2017, date issued: 28 June 2017).

### 2.1. Comparison of Frailty Tests

To compare different established frailty tests and surgical outcome predictors, we also assessed the clinical frailty scale (CFS), timed up-and-go (TUG), 6 min walk test (6 MW), hand grip test (HGT), depression scale (ADS as German version of CESD-Scale), Barthel index, activities of daily living, mini mental test, AGE-Reader, EuroSCORE II, and STS score. We additionally asked for information on accidental falls and the number of hospital stays in the last year. We analyzed correlations with outcomes.

### 2.2. Laboratory Analysis

Routine blood samples before and 1 week after surgery were obtained for laboratory analysis of frailty markers. In-house laboratories analyzed C-reactive protein, vitamin D, hemoglobin, creatinine, urea, NT-Pro-BNP, albumin, HbA1c, cholesterol, high-density lipoprotein, low-density lipoprotein, triglyceride, total protein, interleukin-6, and soluble interleukin-2-receptor. Additionally, proteincarbonyls, malondialdehyde, lutein/zeaxanthin, beta-cryptoxanthine, lycopin, alpha-carotene, beta-carotene, retinol, gamma-tocopherol, and alpha-tocopherol were analyzed externally.

### 2.3. Statistical Analysis

Statistical analyses were performed using R Statistical Software (R Core Team, Vienna, Austria) [27]. Categorical variables are presented as absolute number (n) and percentage. Continuous variables are presented as mean with standard deviation. The chi-square test was used for categorical variables where expected frequencies exceeded 5 in cross-tabulation; otherwise, Fisher’s exact test was used. Group comparisons of normally distributed continuous variables were performed using Student’s unpaired *t*-test for normally distributed variables and the Mann–Whitney U test otherwise. Correlations were compared using Kendall’s TAU. Two-tailed *p*-values of <0.05 were considered to be significant.

## 3. Results

### 3.1. Baseline Characteristics

A total of 250 patients were included pre-operatively, but 4 had to be excluded because they were eventually not operated upon. Sixteen patients (6.5%) were frail and 130 patients (52.85%) were pre-frail, summarized in the frail group (FRAIL) (146 patients) and compared to the non-frail patients (NON-FRAIL). The mean age was 66.5 ± 9.05 years. The percentage that were female was 21.14%. The in-hospital mortality rate was 4.88% and the 1-year mortality rate was 6.1%. A total of 60 patients (24.39%) were >75 years and 8 patients (3.25%) were >80 years. The specific comorbidities and surgical procedures can be seen in Table 1. Comorbidities and Type of Surgery in 3 groups according to fried can be seen in Appendix A.

### 3.2. Frailty Measurements

#### 3.2.1. Frailty Group (Frail)

The main outcome was in-hospital mortality, and the secondary outcomes were length of stay, length of intensive care and intermediate care treatment, deep sternal wound infection, stroke, acute kidney injury, post-operative atrial fibrillation, re-sternotomy due to bleeding, and 1-year all-cause mortality.

FRAIL patients stayed longer in hospital (FRAIL 15.53 ± 8.5 days vs. NON-FRAIL 13.71 ± 8.94 days, *p* = 0.004) and in intensive/intermediate care units (FRAIL 5.4 ± 4.33 days vs. NON-FRAIL 4.86 ± 4.78 days, *p* = 0.014) (Figure 1).

Pre-operatively, they had lower scores in mobility and muscle strength tests such as timed up-and-go (NON-FRAIL 8.81 ± 1.54 sec vs. 10.61 ± 4.07 sec FRAIL *p* < 0.001), 6 min walk (FRAIL 349.14 ± 102.31 vs. 421.54 ± 75.29 m NON-FRAIL l *p* < 0.001), and hand grip strength (NON-FRAIL 39.84 ± 8.35 kg vs. FRAIL 33.33 ± 10.71 kg). This difference could still be seen after surgery (6 MW FRAIL 296.92 ± 100.03 vs. 337.22 ± 87.42 m NON-FRAIL *p* < 0.001, TUG FRAIL 12.31 ± 3.98 vs. 11.07 ± 3.12 sec NON-FRAIL *p* < 0.001, hand grip NON-FRAIL 34.8 ± 8.25 vs. FRAIL 30.25 ± 10.03, *p* < 0.001) (Figure 2). We could not detect significant differences in complications between these two groups (Table 2).

#### 3.2.2. Comparison of Frailty Tests and Risk Scores

1.Mortality: Patients who died during their hospital stay had higher STS scores for mortality, stroke, morbidity or mortality, and long length of stay: Dead vs. survivors in STS mortality: 2.05 ± 1.47 vs. 1.22 ± 1.18, *p* = 0.006. STS stroke: 1.77 ± 1.42 vs. 1.05 ± 0.64, *p* = 0.033. STS morbidity or mortality: 12.59 ± 6.84 vs. 7.84 ± 4.91, *p* = 0.005. STS long length of stay: 7.92 ± 6.17 vs. 4.28 ± 6.85 *p* = 0.004. Regarding 1-year mortality, a longer distance in 6 MW predicted lower mortality (317.92 ± 94.17 m vs. 387.08 ± 93.43 m, *p* = 0.006). Additionally, the MMS scale (25.72 ± 4.36 vs. 27.71 ± 1.9, *p* = 0.048) and clinical frail scale (3.65 ± 1.32 vs. 2.82 ± 0.86 *p* = 0.005) showed differences in 1-year mortality. Unsurprisingly, the EuroSCORE (6.75 ± 9.18 vs. 2.19 ± 1.84, *p* = 0.003) as well as STS scores differed (STS mortality: 2.61 ± 2.09 vs. 1.15 ± 1, *p* < 0.001, STS stroke: 1.69 ± 1.18 vs. 1.03 ± 0.58, *p* = 0.001, STS morbidity and mortality: 13.71 ± 6.87 vs. 7.65 ± 4.66, *p* < 0.001, STS long length of stay: 8.02 ± 5.6 vs. 4.25 ± 7.16, *p* < 0.001).2.Complication rates: Patients with a wound healing disorder had higher STS scores regarding stroke (1.33 ± 0.84 vs. 1.02 ± 0.61, *p* = 0.031) and STS scores regarding long length of stay (4.59 ± 3.6 vs. 4.36 ± 7.13 *p* = 0.038). Patients who suffered from a stroke had a higher EuroSCORE (3.45 ± 2.04 vs. 2.42 ± 2.62, *p* = 0.039) and STS score regarding stroke (1.32 ± 0.47 vs. 1.05 ± 0.65, *p* = 0.048). Additionally, patients who post-operatively presented new episodes of atrial fibrillation had higher STS scores regarding stroke (1.13 ± 0.56 vs. 1.03 ± 0.68, *p* = 0.04). Patients undergoing re-thoracotomy due to bleeding had a higher STS score calculated risk for mortality (1.8 ± 1.36 vs. 1.19 ± 1.13, *p* = 0.028).3.Length of treatment: In-hospital stay correlated with TUG (TAU: 0.094, *p* = 0.037), Barthel index (TAU-0.114, *p* = 0.032), hand grip strength (TAU-0.173, *p* < 0.001), EuroSCORE II (TAU 0.119, *p* = 0.008), STS mortality score (TAU 0.13 *p* = 0.003), STS stroke score (TAU 0.135, *p* = 0.003), STS morbidity or mortality score (TAU 0.106, *p* = 0.017), and STS long length of stay score (TAU 0.098, *p* = 0.027). The duration of intensive care treatment correlated with TUG (TAU 0.196, *p* < 0.001), 6 MW (TAU-0.133, *p* = 0.008), MMS (TAU-0.106, *p* = 0.045), EuroSCORE II (TAU 0.181, *p* < 0.001), STS mortality score (TAU 0.243, *p* <0.001), STS stroke score (TAU 0.171, *p* < 0.001), STS morbidity and mortality score (TAU 0.246, *p* < 0.001), and STS long length of stay score (TAU 0.252, *p* = <0.001). We added an analysis of intensive care treatment including intermediate care treatment. Correlations were found between TUG (TAU 0.186, *p* < 0.001), 6 MW (TAU-0.149, *p* = 0.002), hand grip strength (TAU-0.22, *p* < 0.001), EuroSCORE II (TAU 0.209, *p* < 0.001), STS mortality score (TAU 0.247, *p* < 0.001), STS stroke score (TAU 0.205, *p* < 0.001), STS morbidity and mortality score (TAU 0.21, *p* < 0.001), and STS long length of stay score (TAU 0.232, *p* < 0.001).4.Activities of daily living and AGE-Reader showed no significant correlation with outcome. We had additionally asked for information regarding falls and the number of hospital stays in the last year, which also showed no significant correlation with the outcome of this group.

### 3.3. Laboratory Analysis

Correlations of laboratory markers associated with frailty, in-hospital stay, stay in intensive care unit, and clinical outcome were not significant. We could not find correlations between any laboratory marker with mortality, delirium, renal failure, stroke, pneumonia, new onset of atrial fibrillation, re-thoracotomy or wound healing disorder, hospital stay, or intensive care treatment.

FRAIL and NON-FRAIL patients had post-surgery altered levels of hemoglobin, C-reactive protein, NT-Pro-BNP, albumin, and high-density lipoprotein (Table 3).

FRAIL patients had post-operatively altered levels of plasma redox biomarkers (protein carbonyls, malondialdehyde, and nitrotyrosine) (Figure 3, Table 3) and fat-soluble micronutrients that were also antioxidants (lutein /zeaxanthin, beta-cryptoxanthine, lycopin, alpha-carotene, beta-carotene, retinol, alpha-tocopherol, and gamma-tocopherol) (Figure 4, Table 3). The patients also showed altered levels of creatinine, cholesterol, low-density lipoproteins, and total protein.

For NON-FRAIL patients, there was no significant difference pre- and post-surgery in plasma redox biomarkers or fat-soluble micronutrients. Nevertheless, the overall comparison of FRAIL/NON-FRAIL patients was not significant in all laboratory markers, except for pre-operative vitamin D (49.029 ± 23.628 FRAIL vs. 59 ± 25.053 nmol/L NON-FRAIL, *p* = 0.024). Additionally, the same was found for pre-operative high-density lipoprotein (FRAIL 1.22 ± 0.333 vs. 1.341 ± 0.32 NON-FRAIL mmol/L, *p* = 0.024).

## 4. Discussion

Predicting the risk of poor outcomes after cardiosurgical procedures is essential to assigning individual patients to the most appropriate treatment option, whether that be surgery, intervention, or conservative therapy. However, current risk models for cardiac surgery require some adjustments. Current risk models are based on demographic and clinical factors, as well as comorbidities, but they do not yet consider frailty, which is another important predictor of outcome, particularly in the ever-growing group of elderly patients.

There is growing evidence that frailty assessment tools are valuable in predicting early- and mid-term mortality, in-hospital major adverse events and quality of life following cardiac surgery [17,19,28,29,30,31] It appears that additional frailty assessment scales have better predictive values for adverse outcomes such as delirium in cardiosurgical patients than the EuroSCORE II [32].

There is also growing evidence in patients undergoing transcatheter aortic valve replacement that assessment of frailty can predict postoperative outcomes [33,34,35]. However, the optimal instrument to assess frailty has not yet been determined.

In the absence of a standardized instrument to measure frailty, even screening for components of the frailty-syndrome is worthwhile. Reis et al. showed a correlation between hand-grip-strength and a tendency to fall over as well as mortality [36,37]. In CABG patients the Clinical Frailty Scale, a instrument easily applied in clinical practice, has been shown to predict early and midterm mortality [38]. This could be another option, that can be easily implemented without much effort and showed a significant impact on 1-year mortality in this study. The frailty score as described by Fried (weight loss, handgrip strength, fatigue, 5 m-walking-test and weekly activities) could also be useful because it has a correlation with length of stay and intensive care treatment.

Also, in cardiosurgical patients, a study of Blumenthal et al. also showed an association between depression and mortality [39]. The ADS as the German version of the CESD-Scale did not show a significant correlation in this study but it is only a pure screening tool and does not serve diagnostic purposes.

Afilialo et al. proposed a 4-item scale including lower limb weakness, cognitive impairment, anemia and hypoalbuminemia [40]. We could not find significant results comparing outcome to laboratory markers or correlation to frailty. Before cardiac surgery patients should undergo a comprehensive evaluation addressing all abnormalities carefully. Postoperative levels of micronutrients that serve as antioxidants as well as plasma redox biomarkers and regular laboratory measurements were altered. Diet is a major source of all measured micronutrients. Depletion may thus simply be due to decreased food intake perioperatively. In addition, oxidative stress during surgery and perioperatively may lead to depletion of otherwise adequately available micronutrients simply by overconsumption. Plasma markers indicate oxidative stress following a cardiosurgical procedure. The difference was significant in the frail group, suggesting that they suffered more from the oxidative stress and had fewer reserves.

It is difficult to compare different instruments for measuring frailty, because each of them targets different aspects of frailty syndrome and a single-item laboratory test cannot be the solution. The statistical comparison of these tests, whereby some of which are binary, some are categorical, and some are metric, remains a challenge.

Limitations: As the study is observational in nature, its results depend on the accuracy and completeness of the data collected.

The willingness and ability of patients to participate in our study may have played a role. The incidence of frail patients according to Fried in our cohort was low. Frail patients tend to be referred to a less invasive procedure when advisable and many of them estimate their fitness accurately but do not rate themselves as being suitable for major surgery. They would also not necessarily be attracted to a voluntary study focusing on mobility testing.

## 5. Conclusions

The established risk scores are still necessary and should not be replaced. However, they do not adequately account for frailty. Since frailty causes additional health costs [41] and is an independent predictor of mortality it is very important to know the options and to choose the tests used wisely. Frailty is also a risk factor for long term morbidity and mortality suggesting that it needs to be addressed weather or not a surgery is performed. The current EuroSCORE system and the STS score should therefore be complemented with the most appropriate frailty parameters to further improve the predictive value. Which of the numerous tests should be used is the subject of further studies. Ideally, only a few additional items with good predictive power should be sufficient so that the clinician is not overwhelmed with filling out the form. Unfortunately, laboratory parameters were not sufficient to differentiate between frail and non-frail patients for risk prediction. We suggest further research on the Clinical Frail Scale and the frailty Score by Fried in cardiosurgical patients and possibly incorporate them into scoreing systems in the future.

## Figures and Tables

**Figure 1 jcm-12-03010-f001:**
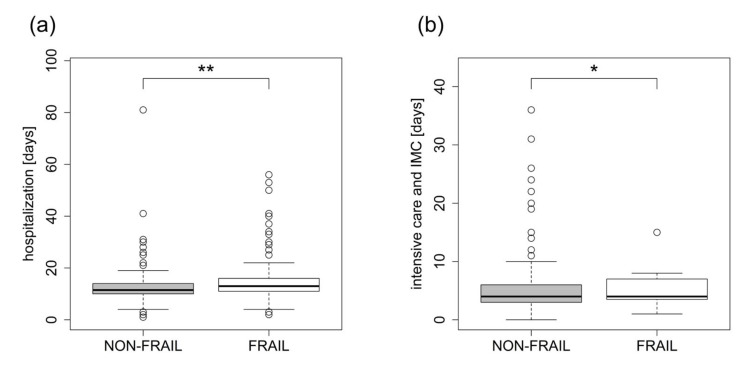
(**a**,**b**) Hospitalization and ICU/IMC stay. In-hospital stay: FRAIL 15.53 ± 8.5 days vs. NON-FRAIL 13.71 ± 8.94 days, *p* = 0.004, and duration of intensive care treatment including intermediate care: FRAIL 5.4 ± 4.33 days vs. 4.86 ± 4.78 days NON-FRAIL, *p* = 0.014. IMC = intermediate care, * *p* < 0.05, ** *p* < 0.01.

**Figure 2 jcm-12-03010-f002:**
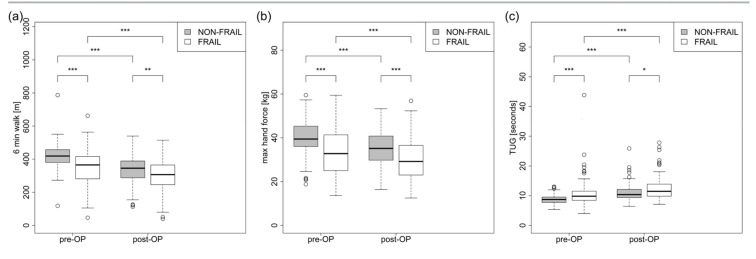
(**a**–**c**) Mobility/muscle strength tests. * *p* < 0.05, ** *p* < 0.01, and *** *p* < 0.001.

**Figure 3 jcm-12-03010-f003:**
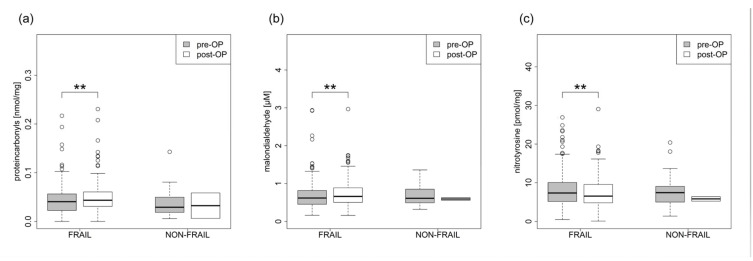
(**a**–**c**) Plasma redox biomarkers. ** *p* < 0.01.

**Figure 4 jcm-12-03010-f004:**
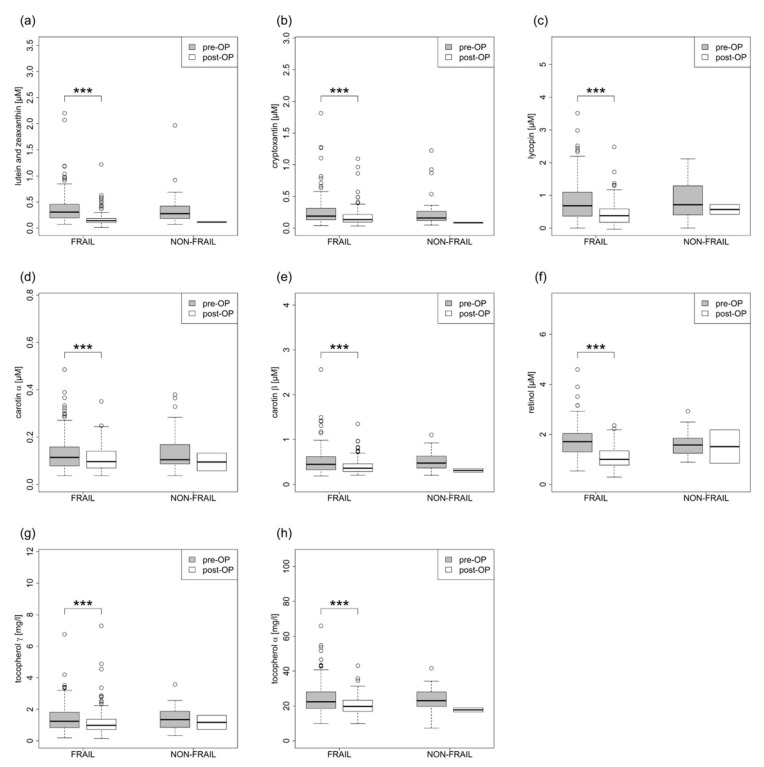
(**a**–**h**): Fat-soluble micronutrients. *** *p* < 0.001.

**Table 1 jcm-12-03010-t001:** Comorbidities and type of surgery.

Comorbidities	All (246)	Frail (146)	Non-Frail (100)	*p*-Value
Age (years)	66.5 ± 9.05	66.68 ± 9.33	66.23 ± 8.67	0.566
Sex (female)	21.14% (52)	25.34% (37)	15% (15)	0.073
Hypertension	72.36%(178)	71.92% (105)	73% (73)	0.967
Diabetes	30.49% (75)	34.93% (51)	24.% (24)	0.091
LVEF	56.86 ± 8.3	56.64 ± 8.94	57.18 ± 7.29	0.914
EuroSCORE II	2.59 ± 3.28	2.94 ± 3.93	2.08 ± 1.92	0.009
BMI (kg/m^2^)	29.71 ± 5.2	29.95 ± 5.57	29.37 ± 4.64	0.712
Surgery				
CABG	40.24% (98)	36.3% (53)	46% (46)	
sAVR	32.11% (79)	31.51% (46)	33% (33)	
Combination CAGB/sAVR	13.41% (33)	13.01% (19)	14% (14)	
Other operation	14.23% (35)	19.18% (28)	7% (7)	

Baseline characteristics: perioperative risk, comorbidities, and complications. LVEF = left ventricular ejection fraction, BMI = body mass index, CABG = coronary artery bypass grafting, sAVR = surgical aortic valve replacement.

**Table 2 jcm-12-03010-t002:** Complications and outcome.

	All (246)	Frail (146)	Non-Frail (100)	*p*-Value
In-hospital mortality	4.88% (12)	4.79% (7)	5% (5)	1
1-year mortality	6.1% (13)	6.45% (8)	5.62 (5)	1
Wound healing disorder	11.48% (28)	11.81% (17)	11% (11)	1
Stroke during stay	3.28% (8)	3.47% (5)	3% (3)	1
Arrythmias	25.61% (63)	26.39% (38)	25% (25)	0.924
Re-thoracotomy	5.69% (14)	5.59% (8)	6% (6)	1
Myocardial infarction	0.82% (2)	0.69% (1)	1% (1)	1
Pneumonia	5.33% (13)	4.86% (7)	6% (6)	0.921
Delirium	10.25% (25)	11.81% (17)	8% (8)	0.454
Intensive/intermediate care	5.18 ± 4.5	5.4 ± 4.33 d	4.86 ± 4.78 d	0.014
In-hospital stay	14.79 ± 8.7 d	15.53 ± 8.5 d	13.71 ± 8.94 d	0.004

**Table 3 jcm-12-03010-t003:** Laboratory analysis of plasma redox biomarkers and fat-soluble micronutrients/antioxidants.

Plasma Redox Biomarkers	Frail Pre-Surgery	Frail Post-Surgery	*p*	Non-Frail Pre-Surgery	Non-Frail Post-Surgery	*p*
Protein carbonyls nmol/mg	0.043 ± 0.03	0.049 ± 0.032	0.003	0.036 ± 0.027	0.032 ± 0.037	1.000
Malondialdehyde µM	0.695 ± 0.394	0.73 ± 0.339	0.009	0.699 ± 0.303	0.594 ± 0.05	0.500
Nitrotyrosine pmol/mg	8.002 ± 4.463	7.309 ± 3.996	0.010	7.881 ± 4.112	5.832 ± 0.834	1.000
Fat-soluble micronutrients/antioxidants						
Lutein / Zeaxanthin µM	0.37 ± 0.272	0.165 ± 0.118	<0.001	0.38 ± 0.341	0.117 ± 0.003	0.500
Beta-Cryptoxanthine µM	0.255 ± 0.215	0.172 ± 0.133	<0.001	0.256 ± 0.26	0.085 ± 0.007	0.500
Lycopin µM	0.805 ± 0.603	0.424 ± 0.339	<0.001	0.877 ± 0.569	0.57 ± 0.217	0.500
Alpha-Carotene µM	0.133 ± 0.077	0.109 ± 0.05	<0.001	0.14 ± 0.088	0.095 ± 0.052	0.500
Beta-Carotene µM	0.508 ± 0.274	0.395 ± 0.155	<0.001	0.517 ± 0.199	0.313 ± 0.056	0.500
Retinol µM	1.719 ± 0.587	1.053 ± 0.392	<0.001	1.58 ± 0.468	1.511 ± 0.945	0.743
Gamma-Tocopherol µM	1.421 ± 0.854	1.155 ± 0.77	<0.001	1.394 ± 0.712	1.168 ± 0.626	0.587
Alpha-Tocopherol µM	24.425 ± 8.548	20.295 ± 5.055	<0.001	23.6 ± 6.751	17.748 ± 1.68	0.205
In-house laboratory						
CRP mg/L	5.555 ± 10.153	84.38 ± 58.67	<0.001	4.241 ± 5.01	63.281 ± 36.178	<0.001
Hemoglobin mmol/L	8.801 ± 0.919	6.49 ± 0.822	<0.001	8.674 ± 1.082	6.374 ± 0.952	<0.001
Creatintine µmol/L	90.09 ± 76.075	81.628 ± 63.889	<0.001	81.235 ± 22.969	86.955 ± 65.741	0.242
Urea mmol/L	6.93 ± 4.34	6.985 ± 4.285	0.400	9.997 ± 19.306	6.375 ± 3.14	0.194
NT-proBNP pg/ml	1123.361 ± 3146.059	2953.946 ± 4782.174	<0.001	1106.085 ± 2729.072	4009.077 ± 9320.021	0.001
Albumin g/L	43.455 ± 3.001	33.812 ± 23.283	<0.001	44.216 ± 6.102	30.667 ± 6.502	<0.001
Cholesterol mmol/L	4.954 ± 1.385	4.043 ± 0.928	<0.001	4.97 ± 0.961	3.873 ± 0.894	0.002
HDL mmol/L	1.22 ± 0.333	0.791 ± 0.17	<0.001	1.341 ± 0.32	0.764 ± 0.174	<0.001
LDL mmol/L	2.802 ± 1.196	2.394 ± 0.755	<0.001	2.911 ± 0.951	2.293 ± 0.78	0.339
Triglyceride mmol/L	2.254 ± 1.637	1.948 ± 0.77	0.083	1.725 ± 0.863	1.774 ± 0.504	0.373
Total protein g/L	70.985 ± 5.075	58.513 ± 6.508	<0.001	69.994 ± 4.737	55.492 ± 9.146	0.003

HDL = high-density lipoprotein, LDL = low-density lipoprotein.

## Data Availability

Data will not be published for privacy reasons, and will be saved at the clinic.

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
