# Peer review of "How Can We Best Measure Frailty in Cardiosurgical Patients?"

_jcm, 2023, doi:10.3390/jcm12083010_

Round 1
Reviewer 1 Report
Thank you for the assignment to review the manuscript entitled " How can we best measure frailty in cardiosurgical patients?". Magdalena L. Laux et al. have performed a prospective observational study of cardiac surgical patients assessing the link between the frailty status (i.e. frail and non-frail) and short and long term outcome. The authors found that frail patients experienced longer ICU/IMC and in-hospital stay. While in-hospital stay correlated with TUG, Barthel Index and handgrip strength, ICU/IMC stay duration showed correlation with TUG, 6MW, and hand grip strength. The authors concluded that "frailty parameters with the highest predictive value as well as ease of use could be added to the EuroSCORE".
While the topic of this investigation is of interest, the manuscript has numerous major concerns regarding the study design, method, and interpretation of results.
Major issues:
1) GENERAL: The authors’ written English requires a native English language editing to achieve more comprehensible text for readers.
2) INTRODUCTION: It is not presented what the clinical perspective and the novelty of this study is. Please try to state a specific hypothesis that the authors aimed to test. Plenty of previous clinical investigations proved that frailty is associated with significantly higher rate of postoperative mortality among cardiosurgical patients. Correspondingly, several recent publications highlighted the importance of assessing frailty before cardiac surgery procedures. Please, see below recent high quality meta-analyses in this field:
Lee, J.A., Yanagawa, B., An, K.R. et al. Frailty and pre-frailty in cardiac surgery: a systematic review and meta-analysis of 66,448 patients. J Cardiothorac Surg 16, 184 (2021). https://doi.org/10.1186/s13019-021-01541-8
Nguyenhuy, M., Chang, J., Xu , R. ., Virk, S. ., & Saxena, A. (2022). The Fried Frailty Phenotype in Patients Undergoing Cardiac Surgery: A Systematic Review and Meta-Analysis. The Heart Surgery Forum, 25(5), E652-E659. https://doi.org/10.1532/hsf.4873
3) METHODS: Regarding the relevant literature of Fried Frailty assessment among cardiosurgical patients, previous studies are divisive in respect of merging or not the pre-frail and frail group of patients. What was the reason to merge these two groups in your statistical analysis? Please, extend section ‘Material and Methods’. Several recent clinical trials proved that the Montreal Cognitive Assessment is prior to the Mini-Mental State Examination in detecting mild cognitive impairment and reflecting cognitive reserve. Furthermore, the primary goal in the development and clinical application of MMSE is dementia screening. The authors should explain the reasons of the MMSE application in their investigation.
RESULTS: The authors should include the data regarding the results of the Fried Frailty assessments. Please, extend the results of your study. The incidence of frailty (excluding the pre-frail patients) among the patients presenting to cardiac surgery in the setting of this research is quite lower, comparing to the results of previous publications. This is an interesting finding and the possible reasons should be explained and discussed. Despite the presented between-group difference in ICU/IMC stay is statistically significant, the 0.54 day (between means) does not seem to be clinically relevant. The importance of this result should be revised.
Reviewer 2 Report
Thank you for the opportunity to review this manuscript.
In this manuscript, the authors performed a prospective observational study with the intent of identifying correlations between preoperative frailty and postoperative outcomes in patients undergoing cardiac surgery. The manuscript is well written, thorough, and describes the frailty measures and outcomes of 246 patients undergoing cardiac surgery.
Comments
1. In the Abstract, the authors state “6-minute-walk (6MW) (317.92 ± 94.17m vs. 387.08 ± 93.43m, p=0.006), Mini Mental Status (MMS) (25.72 ± 4.36 vs. 27.71 ± 1.9, p=0.048) and Clinical Frail Scale (3.65 ± 1.32 vs. 2.82 ± 0.86 p=0.005) showed different 1-year mortality.” It is unclear what is meant by the 6MW, MMS, and Clinical Frail Scale showed different 1-year mortality as mortality rates were not reported in this sentence. This sentence should be revised for clarity.
2. While the authors listed a number of assessments of frailty that were used in this study, it is not clearly defined how patients were assigned to each group. How many indicators of frailty were required for a patient to be assigned to the pre-frail or frail group? This should be described.
3. Has there been any consideration for preoperative optimization of patients identified to be frail? While percutaneous therapies such as PCI and TAVR may be options for patients at high or prohibitive surgical risk, many patients will still require or benefit from surgery, even if found to be frail. The identification of frail patients may allow for preoperative optimization, time permitting, of nutritional status and improvement of physical conditioning. It would be interesting if the authors were to discuss the identification of frailty preoperatively to inform potential preoperative optimization of patients. This is commonly used in heart and lung transplantation and may be of use in the frail cardiac surgical patient as well.
4. The tables and figures provided are appropriate and complement the information provided in this study.
Round 2
Reviewer 2 Report
Thank you for the opportunity to review this manuscript.
In this manuscript, the authors performed a prospective observational study with the intent of identifying correlations between preoperative frailty and postoperative outcomes in patients undergoing cardiac surgery. The manuscript is well written, thorough, and describes the frailty measures and outcomes of 246 patients undergoing cardiac surgery.
Comments
1. The authors have successfully addressed my previous comments. I have no further suggestions for improvement.